# Genome-Wide Identification, Characterization, and Expression Analysis of DDE_Tnp_4 Family Genes in *Eriocheir sinensis*

**DOI:** 10.3390/antibiotics10121430

**Published:** 2021-11-23

**Authors:** Yuanfeng Xu, Jinbin Zheng, Yanan Yang, Zhaoxia Cui

**Affiliations:** 1School of Marine Sciences, Ningbo University, Ningbo 315020, China; 1901130057@nbu.edu.cn (Y.X.); zhengjinbin@nbu.edu.cn (J.Z.); cuizhaoxiao@nbu.edu.cn (Z.C.); 2Laboratory for Marine Biology and Biotechnology, Qingdao National Laboratory for Marine Science and Technology, Qingdao 266071, China

**Keywords:** DDE_Tnp_4 family, *Eriocheir sinensis*, gene expansion, environmental stress

## Abstract

DDE transposase 4 (DDE_Tnp_4) family is a large endonuclease family involved in a wide variety of biological processes. However, little information is available about this family in crustaceans. In this study, we used HMMER to identify 39 DDE_Tnp_4 family genes in *Eriocheir sinensis* genome, and the genes were classified into four subfamilies according to phylogenetic analysis. Gene expansions occurred among *E. sinensis* genome, and synteny analysis revealed that some DDE_Tnp_4 family genes were caused by tandem duplication. In addition, the expression profiles of DDE_Tnp_4 family genes in *E. sinensis* indicated that subfamily I and II genes were up-regulated in response to acute high salinity and air exposure stress. *E. sinensis* is a kind of economical crustacean with strong tolerance to environmental stress. We confirmed the expansion of DDE_Tnp_4 family genes in *E. sinensis* and speculated that this expansion is associated with strong tolerance of *E. sinensis*. This study sheds light on characterizations and expression profiles of DDE_Tnp_4 family genes in *E. sinensis* and provides an integrated framework for further investigation on environmental adaptive functions of DDE_Tnp_4 family in crustaceans.

## 1. Introduction

The Chinese mitten crab, *Eriocheir sinensis,* is one of the most commercially important crustaceans [1]. The life history of *E. sinensis* is more complicated than other economic crabs due to a special breeding migration. Adult *E. sinensis* mostly live in freshwater areas far from estuaries [2]. During breeding season, *E. sinensis* migrates downstream from fresh to brackish water, where they suffer complicated stress [3]. *E. sinensis* is a euryhaline species and a strong osmoregulator as they can function equally well in freshwater or brackish environments [4]. In addition, *E. sinensis* is a species capable of aerial respiration and can survive for extended periods of time, even days, without water [5]. Compared with *Scylla paramamosain* and *Portunus trituberculatus*, which are also commercially important crustaceans in China, *E. sinensis* exhibits enhanced tolerance to abiotic factors, including high salinity and air exposure [2,5,6,7]. Existing studies have revealed that strong tolerance to high salinity and air exposure of *E. sinensis* is linked to genes participating in innate immune system and many other physiological activities [8,9], such as Toll-like receptor (TLR), Na^+^-K^+^-ATPase, heat shock proteins (like Hsp70 and Hsp90), and superoxide dismutase (SOD) [10,11,12].

Furthermore, gene expansion has been associated with environmental adaptation [13]. In *Chlamys farreri*, the expansive Cu/Zn SOD family genes are considered to assist in protecting the body against paralytic shellfish toxins (PSTs) [14]. The expansive subfamilies G and H of ATP-binding cassette (ABC) family in *Daphnia pulex* are thought to be critical in pollutant efflux and cell defense activities [15]. However, whether excellent abiotic stress tolerance of *E. sinensis* is linked to some expansive genes in *E. sinensis* remains unknown.

DDE superfamily is a huge endonuclease family characterized by the presence of three conserved acidic residues (Asp-Asp-Glu) in their RNase H–like domain (RNH) active sites that can bind to magnesium ion and catalyze phosphodiester bond hydrolysis [16,17]. DDE transposase 4 (DDE_Tnp_4) family, belonging to DDE superfamily, is required for efficient DNA transposition. Some DDE_Tnp_4 family genes function as transcription factors, regulating the expression of diverse sets of genes implicated in immune regulation, angiogenesis, cell cycle regulation, stem cell pluripotency, and epigenetic gene silencing [18,19,20]. 

DDE_Tnp_4 family genes have been reported to expand during the evolution of some species, providing them with some evolutionary advantages. One member of DDE_Tnp_4 family is costly and exhibits evolutionary conservation patterns in coelacanth, *Latimeria chalumnae* genome, which benefits vertebrate invasion of terrestrial environment [21]. Several DDE_Tnp_4 family genes have undergone differential patterns of lineage-specific expansion in *Acyrthosiphon pisum*, implying that these genes might have a significant role in the diversification of morphology and adaptations [22]. 

This study was conducted to identify whether *E. sinensis* DDE_Tnp_4 family has expanded and whether this family is linked to stress tolerance of *E. sinensis*. We first identified and compared DDE_Tnp_4 family genes among different genomes in crustaceans and then investigated the localization of DDE_Tnp_4 family genes on *E. sinensis* chromosomes. Transcriptome data were employed to analyze the expression changes of *E. sinensis* DDE_Tnp_4 family genes under high salinity and air exposure stress. This study provides the basic information and clues for further investigation of the environmental adaptive function of DDE_Tnp_4 family.

## 2. Results

### 2.1. The Expansion of DDE_Tnp_4 Family

Herein, 39, 16, 3, and 4 DDE_Tnp_4 family genes were identified in *E. sinensis*, *S. paramamosain*, *P. trituberculatus,* and *D. pulex*, respectively. Meanwhile, the number of DDE_Tnp_4 family genes accounted for 0.1391%, 0.0872%, 0.0072%, and 0.0131% of whole-genome protein-coding genes in *E. sinensis*, *S. paramamosain*, *P. trituberculatus,* and *D. pulex*, respectively (Amino acid sequences can be found in Appendix A). The computational analysis of gene family evolution (CAFE) output result indicated that, among the three Brachyura species tested, DDE_Tnp_4 family genes of *E. sinensis* branch expanded significantly (*p* = 1.97756 × 10^−6^) compared with Brachyura ancestor species (Figure 1), with a total of 20 expansions. This result indicated that DDE_Tnp_4 family expanded among *E. sinensis* genome.

Phylogenetic analysis revealed that DDE_Tnp_4 family genes could be categorized into four groups: I, II, III, and IV subfamilies (Figure 2). About 23 DDE_Tnp_4 family genes, including 14 from *E. sinensis*, 7 from *S. paramamosain,* and 4 from *D. pulex*, were clustered in subfamily I. A total of 18 DDE_Tnp_4 genes, including 17 from *E. sinensis* and 1 from *P. trituberculatus*, were clustered in subfamily II. Nine DDE_Tnp_4 genes, including seven from *E. sinensis* and two from *S. paramamosain*, were clustered in subfamily III. In the end, ten DDE_Tnp_4 genes, including one from *E. sinensis*, seven from *S. paramamosain,* and two from *P. trituberculatus*, were clustered in subfamily IV (Figure 2). The phylogenetic tree confirmed that DDE_Tnp_4 family was widely present in crustaceans, and there were certain differences in this family among different species.

### 2.2. Conserved Motif and Gene Structure of DDE_Tnp_4 Family Genes in E. sinensis

Multiple Em for Motif Elicitation (MEME) software was used to identify 15 conserved motifs among these *E. sinensis* DDE_Tnp_4 family genes. Each subfamily had its own motif distribution mode, and some motifs only appeared in a special subfamily (Figure 3). Motif 4 existed in all subfamily II genes and a part of subfamily I genes, Motif 1 was found in almost all subfamily II and III genes, Motif 6 was only present in a part of subfamily II genes, and Motif 15 was only found in subfamily III genes. This result indicated that different subfamilies had their own MEME characteristics, confirming the classification of the subfamilies of *E. sinensis* DDE_Tnp_4 family.

According to whole-genome gene annotations, the gene structure of *E. sinensis* DDE_Tnp_4 family genes was analyzed. Among 39 identified genes, 43.59% (17) of genes did not have introns, 33.33% (13) owned just one intron, 20.51% (8) possessed two introns, and only 2.56% (1) had three introns (Figure 4). In addition, lengths of these introns were found to exhibit polymorphism. Most lengths of these introns (93.94%) varied from 100 to 3000 base pairs (bp). However, a part of introns (6.06%) was quite long, exceeding 10,000 bp in length, with the longest reaching 12,791 bp (one intron of CCG071364.1).

### 2.3. Synteny and Duplication of DDE_Tnp_4 Family Genes in E. sinensis

In total, 39 DDE_Tnp_4 family genes were mapped to 28 chromosomes of *E. sinensis* genome, and each chromosome contained 1–3 DDE_Tnp_4 family genes (Appendix A). However, there were three couples of genes: (CCG011549.1 and CCG011535.1, CCG036054.1 and CCG036055.1, CCG037687.1 and CCG037686.1) tightly linked to each other, and their protein sequences were relatively conserved (Figure 5).

### 2.4. Gene Expression of E. sinensis DDE_Tnp_4 Family Genes under Acute High Salinity Stress and Air Exposure Stress

To investigate the potential function of *E. sinensis* DDE_Tnp_4 family genes in coping with environmental stress, *E. sinensis* transcriptomes under acute high salinity and air exposure stress conditions were analyzed, the details of expression analysis, clustering analysis, and normalization information can be found in Appendix A. Under acute high salinity stress conditions, 14 DDE_Tnp_4 genes were found to be expressed in *E. sinensis* hemocytes, among which 8, 4, 1, and 1 genes belong to subfamily I, II, III, and IV, respectively. Under air exposure stress conditions, 12 DDE_Tnp_4 genes were found to be expressed in gills of *E. sinensis*, among which 7, 3, 1, and 1 genes belong to subfamily I, II, III, and IV, respectively. 

Under acute high salinity stress conditions, the expression of subfamily I genes showed largely high expression compared to control groups, while subfamily II genes were also up-regulated. Under air exposure stress conditions, subfamily I, II, and IV genes were generally up-regulated (Figure 6). This result indicated that DDE_Tnp_4 family might be involved in high salinity and air exposure stress conditions in *E. sinensis*.

## 3. Discussion

DDE_Tnp_4 is a large family containing numerous members which function in angiogenesis, cell cycle regulation, and innate immunity [18,19,20]. Herein, we identified 39 DDE_Tnp_4 family genes in *E. sinensis* genome, which are dispersedly distributed across 28 chromosomes. This phenomenon is consistent with the expansion of transposon-derived Iris genes in *Drosophila melanogaster* through “cut and paste” mechanism [23]. However, three couples of subfamilies II and III genes were found to be tightly linked to each other. According to judgment standards of tandem genes [24], these three couples of genes could be identified as tandem genes and resulted from tandem duplication, similar to numerous TLR genes found in *Strongylocentrotus purpuratus* [25]. Therefore, we speculate that DDE_Tnp_4 family genes in *E. sinensis* may be formed by various mechanisms such as “cut and paste” mechanism and tandem duplication of transposable elements. The number of introns in *E. sinensis* DDE_Tnp_4 family genes is relatively small; 43.59% (17) of genes lacked having introns, while 33.33% (13) of genes owned just one intron. This result is similar to the situation found in *Danio rerio*, where DDE_Tnp_4 family genes found in *D. rerio* lacked or possessed only one intron [26].

Transposons are a significant basic source of gene expansion. During species evolution, transposons replicate, move, amplify, and accumulate in invaded genomes [27]. Following molecular domestication, these inserted transposons become novel functional genes in host genomes [27]. To determine whether DDE_Tnp_4 family genes in *E. sinensis* have expanded, the genomic information of *E. sinensis* was compared with three other crustaceans. It was discovered that *E. sinensis* had a greater number and ratio of DDE_Tnp_4 family genes than *S. paramamosain*, *P. trituberculatus,* and *D*. *pulex*. Furthermore, expansion and contraction analyses confirmed that DDE_Tnp_4 family in *E. sinensis* genome expanded compared with other Brachyura species. The above findings are consistent with those reported for *L. chalumnae*; the expansion of DDE_Tnp_4 family genes is observed, and this phenomenon is speculated to be linked to adaptation of aquatic organisms to terrestrial environment [21]. Expansion of DDE_Tnp_4 family genes is also observed in *A. pisum*, and these expansive genes might have a significant role in the diversification of morphology and adaptations [22].

During gene expansion, the positive selection pressure promotes species to form more novel functional genes, which is more conducive for species to adapt to various biological or abiotic stresses [13]. For instance, Cu/Zn SOD family genes in *C. farreri* have expanded significantly, which protects the body against the damaging effects from PSTs [14]. There are gene expansion phenomena among subfamilies G and H of ABC family in *D*. *pulex*, and these genes are involved in pollutant efflux and cell defense activities [15]. *E. sinensis* has complicated life history and high tolerance to environmental stress [3]. To verify whether excellent environmental adaptability of *E. sinensis* is linked to expansive DDE_Tnp_4 family genes, we analyzed expression profiles of *E. sinensis* DDE_Tnp_4 family genes under acute high salinity and air exposure stress conditions. Most DDE_Tnp_4 family genes were generally up-regulated under acute high salinity and air exposure stress conditions. The gene expression of subfamily I under high salt stress conditions was largely up-regulated compared to the control group. The aforementioned findings indicated that DDE_Tnp_4 family might be involved in high salinity and air exposure stress. These results are consistent with those obtained in *Marsupenaeus japonicus*, where the expression of DDE_Tnp_4 family genes was significantly up-regulated when challenged with white spot syndrome virus (WSSV) [20].

According to the above results, it is speculated that expensive DDE_Tnp_4 family genes may be linked to environmental tolerance of *E. sinensis*. In addition, *E. sinensis* could adapt to abiotic stresses by activating innate immune system and many other physiological activities [8,9]. High salinity and air exposure stress have been demonstrated to activate TLR pathway and antioxidant response in *E. sinensis* [10,11,12]. As transcriptional regulators, DDE_Tnp_4 family is considered to specifically regulate transcription of target genes [26]. For instance, DDE_Tnp_4 family genes participate in TLR pathway by positively regulating Toll genes expression in *M. japonicus* [20]. As a result, we speculated that DDE_Tnp_4 family genes might participate in *E. sinensis* adaptation to environmental stress by regulating the transcription of target genes in immune system and physiological activities.

## 4. Materials and Methods

### 4.1. Identification of DDE_Tnp_4 Family Genes

The *E. sinensis* genome and annotation data were obtained from our previous study (accession number: LQIF00000000). The genome data of *S. paramamosain*, *P. trituberculatus,* and *D*. *pulex* were downloaded from the National Center for Biotechnology Information (NCBI) database (accession number: SRR12442560, SRR12442561, GCA_008373055.1, and GCA_000187875.1). To identify DDE_Tnp_4 family genes, Hidden Markov Model (HMM) profile of DDE_Tnp_4 family genes (accession number: PF13359) was downloaded from Pfam database [28], and a set of proteins from the genome of *E. sinensis* was scanned using hmmsearch from HMMER v3.3 suite [29] using HMM with an expected value threshold < 10^−4^ [30]. The results were further confirmed using the online tool SMART [31] (http://smart.embl.de/, accessed on 8 October 2021). DDE_Tnp_4 family genes of *S. paramamosain, P. trituberculatus,* and *D*. *pulex* were also identified using the same methods.

### 4.2. Gene Family Expansion and Contraction Analysis

OrthoFinder [32] was used to identify single copy orthologue genes among genome data of *E. sinensis*, *S. paramamosain, P. trituberculatus,* and *D*. *pulex*. MEGA-X program was employed to construct the ultrametric tree according to single copy orthologue genes and fossil information from Fossil Calibration Database [33,34]. The numbers of DDE_Tnp_4 family genes and the ultrametric tree were input into CAFE to analyze the expansion and contraction situation. The output result was visualized using Interactive Tree of Life (iTol) [35] (https://itol.embl.de/, accessed on 8 October 2021).

### 4.3. Gene Family Expansion and Contraction Analysis

Multiple sequence alignments were performed using MUSCLE tool [36]. A Maximum Likelihood (ML) phylogenetic tree was constructed by MEGA-X program using General Time Reversible (GTR) model with 1000 bootstrap replicates [33], and iTol [35] (https://itol.embl.de/, accessed on 8 October 2021) was used to display the phylogenetic tree generated from MEGA-X program.

### 4.4. Localization and Synteny Analysis of DDE_Tnp_4 Family Genes

Location and synteny information of DDE_Tnp_4 family genes were obtained from *E. sinensis* genome annotations. Gene Location Visualize (Advanced) [37] was used to display DDE_Tnp_4 family genes on *E. sinensis* chromosomes. Tandem genes in *E. sinensis* were identified as previously reported standards [24]. Multiple sequence alignment of proteins was performed using CLUSTALW program packaged in DNAMAN 8.0 software.

### 4.5. Localization and Synteny Analysis of DDE_Tnp_4 Family Genes

MEME 5.1.1 [38] (http://meme-suite.org/tools/meme, accessed on 28 September 2021) was used to scan conserved motifs. Parameters in MEME were as follows: the number of motifs, 15; minimum width, 6; maximum width, 200; other parameters were left at their default values. The global perspective of motifs in each DDE_Tnp_4 family genes was conducted by HeatMap [37]. The Gene Structure View (Advanced) [37] was employed to display the gene exon-intron structure and domain coding regions within default parameters.

### 4.6. Expression Profiling of DDE_Tnp_4 Family Genes under Stress

Under stress conditions, the profiles of DDE_Tnp_4 family genes were analyzed using transcriptome data obtained from previous studies [5] and the European Nucleotide Archive (ENA) database. During acute high salinity stress experiment, healthy crab individuals were divided into four groups; one group in freshwater serves as the control, and other three groups were challenged with 16‰, 28‰, and 35‰ saltwater, respectively. Following that, hemocytes were sampled for RNA extraction at 24 h postsalinity challenge. During the air exposure stress experiment, healthy crab individuals were cultured for 1, 3, and 5 days with constant 27 °C, 95% humidity, 14 h light, 10 h dark, and no water. For the control group, ten crab individuals were cultured in water (27 °C). Then, gill tissues from each group were quickly collected and used to extract RNA. Raw data of transcriptome sequencing were downloaded from ENA database (accession number: SRR6516036, SRR6516037, SRR6516038, SRR6516039, SRR6516040, SRR6516041, SRR7507779, SRR7507780, SRR7507781, and SRR7507782). Quality control and filtering of raw reads were conducted using FastQC [39] and Trimmomatic [40] software, respectively. De novo transcriptome assembly was performed using Trinity 2.6.6 [41]. Expression abundance of transcript was calculated as transcripts per million mapped reads (TPM) value by using scripts in Trinity. The read counts were adjusted by edgeR program package through one scaling normalized factor, and differential expression analysis of two samples was performed using DEGseq R package [42]. The global perspective of DDE_Tnp_4 family genes expression level was conducted using ggplot2 R package [43].

## Figures and Tables

**Figure 1 antibiotics-10-01430-f001:**
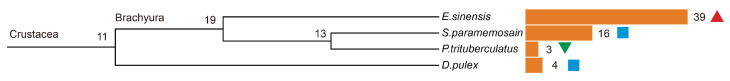
Distribution of DDE transposase 4 (DDE_Tnp_4) family genes gain and loss among crustaceans. The numbers on nodes represent a total number of DDE_Tnp_4 family genes for ancestral species. The red positive triangle represented that genes in this species expanded significantly. The green inverted triangle represented that genes in this species contracted significantly. The blue square represented that genes in this species did not expand or contract significantly.

**Figure 2 antibiotics-10-01430-f002:**
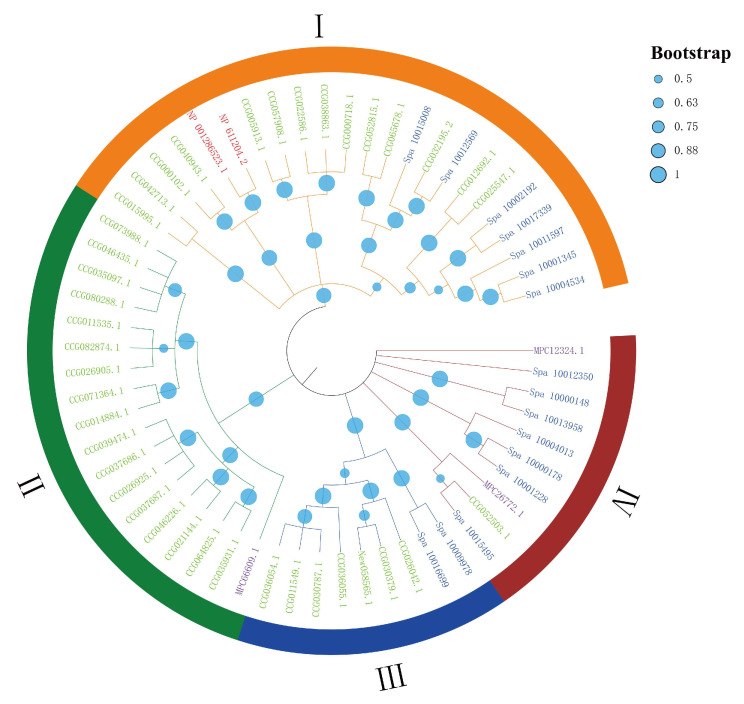
The phylogenetic tree of DDE_Tnp_4 family genes. *Eriocheir sinensis*, *Scylla paramamosain*, *Portunus trituberculatus,* and *Daphnia pulex* genes were distinguished by green, blue, purple, and red, respectively. Subfamily I, II, III, and IV were highlighted in orange, green, dark blue, and dark red, respectively.

**Figure 3 antibiotics-10-01430-f003:**
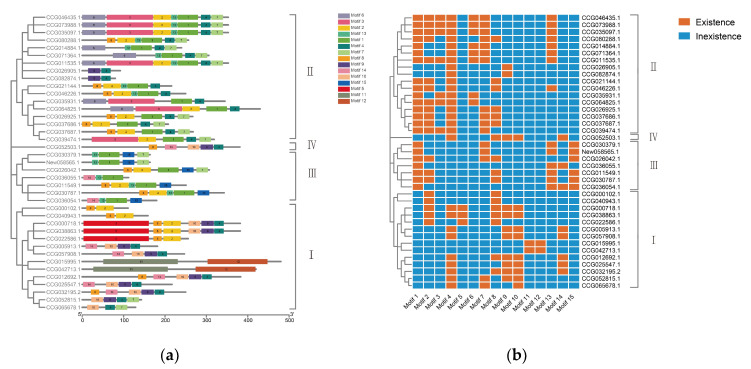
Conserved motifs of DDE_Tnp_4 family genes in *E. sinensis*. (**a**) The architecture of conserved protein motifs. (**b**) The heatmap indicates whether the meme motif existed in this gene. The orange modules imply that this gene contains this motif, while the blue ones do not.

**Figure 4 antibiotics-10-01430-f004:**
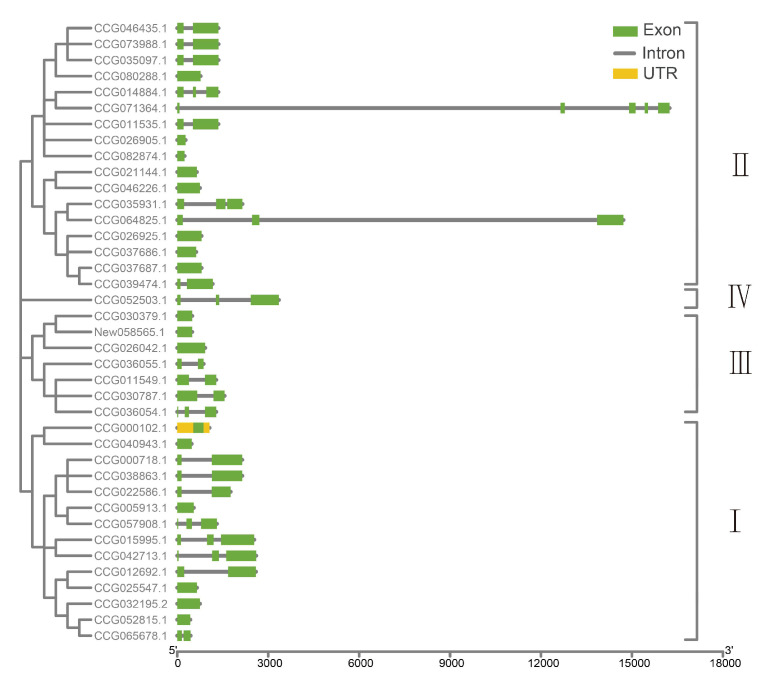
Gene structures of DDE_Tnp_4 family genes in *E. sinensis*. The green models mean exons, yellow models mean untranslated regions (UTR), and grey lines mean introns.

**Figure 5 antibiotics-10-01430-f005:**
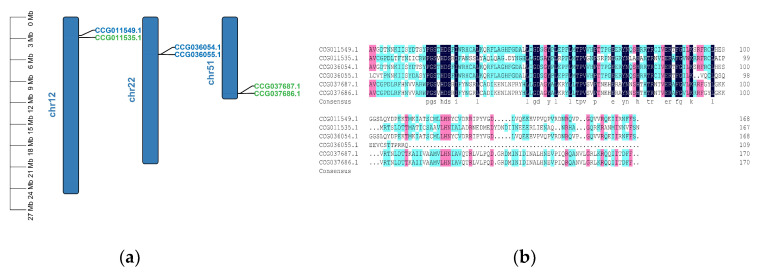
Tightly linked DDE_Tnp_4 family genes in *E. sinensis*. (**a**) Distribution of these genes identified in *E. sinensis* chromosomes. Subfamily II and III genes were distinguished by green and blue, respectively. (**b**) Multiple alignments of protein sequences. The dark blue shading indicates identical amino acid residues. Pink shading indicates less conserved residues. Light blue shading indicates somewhat similar residues.

**Figure 6 antibiotics-10-01430-f006:**
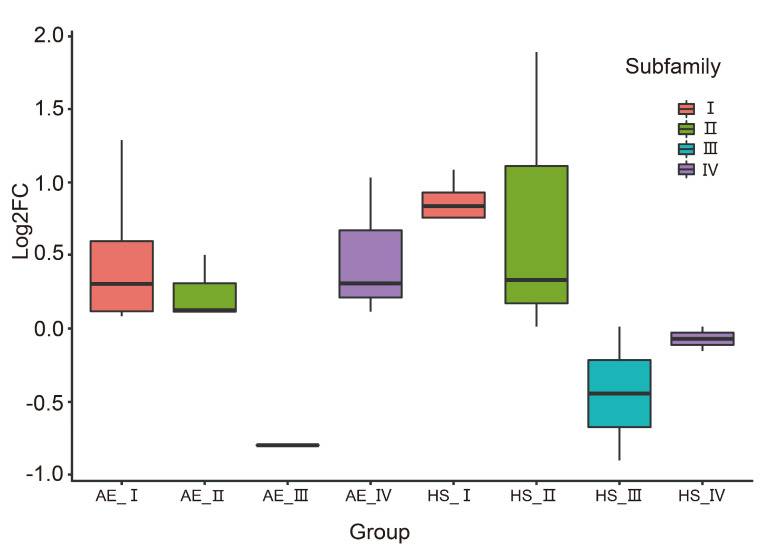
Global expression profiles of DDE_Tnp_4 family genes in *E. sinensis* hemocytes under acute high salinity conditions and in *E. sinensis* gills under air exposure stress conditions. AE_I-AE_IV represents respective subfamily genes under air exposure stress conditions. HS_I-HS_IV represents respective subfamily genes under acute high salinity stress conditions.

## Data Availability

The data presented in this study are available in Appendix A.

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
