# Peer review of "Genome-Wide Identification, Characterization, and Expression Analysis of DDE_Tnp_4 Family Genes in Eriocheir sinensis"

_antibiotics, 2021, doi:10.3390/antibiotics10121430_

Round 1
Reviewer 1 Report
This manuscript entitled “Genome-wide identification, characterization and expression analysis of DDE_Tnp_4 family genes in Eriocheir sinensis” is observational as the authors lack the experiments to validate the observation. The authors identified the DDE transposase 4 (DDE_Tnp_4) family genes and predicted their roles in diversification of morphology and adaptations to E. sinensis. This manuscript may encourage the researchers to explore how crustaceans have strong tolerance. Below are my comments:
Major comments:
- Authors should perform qRT-PCR (quantitative real-time PCR) for confirming the reliability of the RNA-Seq results.
- All figures should be replaced with the corresponding high dpi images because the images are not clear.
- Authors should show data normalization and distribution in the supplementary figures.
Author Response
Thank you for your review work. Please see the attachment.

Reviewer 2 Report
Dear authors,
The paper analyses of characterization and expression of DDE_Tnp_4 family genes in Eriocheir sinensis. The manuscript is well written and structured. The introduction provides sufficient background and include all relevant references, the research design is appropriate, the methods are adequately described, the results are clearly presented, and the conclusions are supported by the results. The manuscript can be published in present form.
The manuscript addresses the study of the DDE Tnp 4 gene family in the crustacean specie Eriocheir sinensis through a complete genome analysis, in order to relate this gene family to tolerance to stress due to salinity and exposure to air.
The topic is original because it is the first time that genes related to stress resistance in crustaceans have been studied. In addition, the species studied in particular has an important economic interest.
To the best of my knowledge, this manuscript uses a whole genome analysis to address stress resistance in crustaceans.
In my opinion, the methodology is adequate for the objective set out in the work.
The conclusions are consistent with the Results obtained in This work.
The references are appropriate.
The tables and figures on the manuscript are clearly presented and greatly facilitate the understanding of the study.
Author Response

(The authors gave the same response as above.)

Reviewer 3 Report
In this manuscript, Xu et al. conducted in silico analyses of the DDE_Tnp_4 family genes in an economically important crab species. While some people may claim that this is merely a conventional cloning-expression analysis study, I consider that this manuscript is worth to be published because it can be a new standard that replaces traditional cloning studies. All methods are scientifically sound and well described. All of my comments are minor.
L27: Please add details; "more complicated" compared to what?
L38: Consider rephrasing. "innate immune system many other physiological activities" does not match with the latter part of the sentence. "Toll-like receptor..." are proteins. Some of them are not immune system nor activities. Also, the authors are citing HSP70 papers here, but this requires careful annotation. HSP70 classification is being changed. For details, see:
Scientific Reports (2021) 11:17794
Comparative Biochemistry and Physiology, Part A (2021) 262, 111060
Fig. 3: Subfamily IV contains only one gene, and it is difficult to tell whether the feature we observe in Fig. 3 is a common characteristic of this subfamily. I suggest to make a similar figure (supplementary) including all genes shown in Fig. 2.
Fig. 6: I am not sure how informative this data is because genes in these subfamilies can be regulated differently. At least row expression analysis (like a heatmap including expression data of all genes) should be provided as a figure or a supplementary figure. Clustering analysis of the row data should also be informative.
L145, Fig. 6: Please clarify which tissue was used.
Author Response

(The authors gave the same response as above.)

Round 2
Reviewer 1 Report
Comments have been addressed and authors have made necessary modifications needed in the manuscript.